# Exploring mechanisms of Neural Robustness: probing the bridge between geometry and spectrum

## Abstract

Backpropagation-optimized artificial neural networks, while precise, lack robustness, leading to unforeseen behaviors that affect their safety. Biological neural systems do solve some of these issues already. Thus, understanding the biological mechanisms of robustness is an important step towards building trustworthy and safe systems. Unlike artificial models, biological neurons adjust connectivity based on neighboring cell activity. Robustness in neural representations is hypothesized to correlate with the smoothness of the encoding manifold. Recent work suggests power law covariance spectra, which were observed studying the primary visual cortex of mice, to be indicative of a balanced trade-off between accuracy and robustness in representations. Here, we show that unsupervised local learning models with winner takes all dynamics learn such power law representations, providing upcoming studies a mechanistic model with that characteristic. Our research aims to understand the interplay between geometry, spectral properties, robustness, and expressivity in neural representations. Hence, we study the link between representation smoothness and spectrum by using weight, Jacobian and spectral regularization while assessing performance and adversarial robustness. Our work serves as a foundation for future research into the mechanisms underlying power law spectra and optimally smooth encodings in both biological and artificial systems. The insights gained may elucidate the mechanisms that realize robust neural networks in mammalian brains and inform the development of more stable and reliable artificial systems.

## 1 Introduction

Research on Artificial Intelligence (AI) has made tremendous progress within recent decades. To a large extent, this success is due to biologically inspired artificial neural networks (ANN) with vast parameter spaces. Convolutional Neural Networks (CNNs) constitute a prominent example. Mimicking the morphology of the visual cortex, they have revolutionized the field of image analysis Krizhevsky et al. (2012). Today, the majority of ANNs are trained supervised using backpropagation.

Despite achieving high accuracy, backpropagation optimised ANNs are unstable with regard to changes in their input Antun et al. (2020). For example, unpredictable changes in output are caused by random noise or adversarial examples on the input Goodfellow et al. (2015). Instabilities can lead to unexpected model behaviors with direct consequences for the applicability of AI technology. In cancer recognition, for example, hardly visible changes in images of moles can cause diagnostic tools to change their rating from benign to malignant Finlayson et al. (2019), at the expense of the patient's health. Not only does this example illustrate direct individual implications, but it also questions the reliability of such systems and can be a problem of societal impact.

When compared to humans in a feed forward setting, machine learners are significantly less robustness against black box attacks Geirhos et al. (2018). This type of divergence is hypothesized to be due to invariances in model "metamers" compared between biological and artificial neural networks Feather et al. (2023) and perceptual straightness of visual representations Harrington et al. (2022). However, when compared to primates, black box attacks show that inferior temporal gyrus neurons

are more suceptible than adverserially trained networks Guo et al. (2022). Out-of-distribution (ood) generalization compared between humans and machine learners shows that the primary factors for increasing robustness are data size and architectural design Geirhos et al. (2021). This example illustrates that it is promising to search for properties and mechanisms in biological systems that ANNs might benefit from and vice versa. Biological Neural Networks (BNN) are models that emulate the nature of neural tissue beyond a connected set of neurons. Moreover, BNNs adjust their connectivity in response to the activity patterns of neighboring neurons within the network. In that sense, BNNs learn locally. A prominent example of a local learning algorithm is Oja's rule Oja (1982) which is a mathematical formalisation of Hebb's learning theory Hebb (1949). Since BNNs learn differently from ANNs, implementing principles such as local learning constitutes one potential approach to resolve robustness issues.

Recent work from Krotov & Hopfield (2019) introduced the idea to learn latent representations using a biologically plausible local learning rule in an otherwise backpropagation optimized model. Grinberg et al. (2019) demonstrated that models that feature such biologically plausible layers can at least keep up with end-to-end backpropagation trained networks in terms of accuracy. Additionally, Patel & Kozma (2020) showed similar models to be more resilient to black box attacks, such as square occlusion, than their end-to-end counterparts. Black box attacks are perturbation methods that treat models as black boxes. In contrast to them, white box attacks have access to the inner workings of a model and can, therefore, fool them more specifically. These studies find that local learning yields smoother feature maps compared with their end-to-end counterparts, and conclude that to be the reason for the observed increase in robustness.

Representations on smoother manifolds are less affected by small perturbations in the input which makes them less prone. From a geometric perspective, a representation's smoothness relates to how abruptly the phase space surface it spans passes over from one point to another. However, smaller distances between closely related representations makes them also harder to distinguish. Therefore, different representations of data are usually pushed apart to increase expressivity. Thus, from a structural perspective, accuracy and robustness stay in an inverse relationship and are conflicting. Stringer et al. (2019) proved that a representation's fractal dimension, which can be considered a measure of smoothness, is related to the exponent of asymptotic decay in the manifold's covariance (PCA) spectrum. As a consequence, an optimal balance between accuracy and robustness is characterized by a close to $n^{-\alpha}$ power law decay in ordered spectral components, where $\alpha$ depends on the input's intrinsic dimension. Interestingly, it is this power law functional relation that they also observe in the primary visual cortex of mice. Not only does this result validate their argument, but it also suggests V1 representations to be optimal in that sense. In consequence, their study indicates that instabilities of artificial neural network models may be related to the smoothness of their representations, as compared to biological neural networks.Stringer et al. (2019) Following this, Nassar et al. (2020) introduced a power law spectral regularization term to enforce their image classifiers to favor power law representations in their hidden layers under supervised learning. In agreement with Stringer, they found representations following a power law to be more robust in Multi-Layer Perceptron Models (MLPs) and CNNs. However, their results are solely empirical and the underlying mechanisms are not understood.

Instead of relying on empirical evidence linking the spectrum to robustness, one can optimize for smooth representations directly. Assuming the representation's phase surface to be locally differentiable, the norm of its Jacobian constitutes a valid local measure of change and, hence, smoothness. Thus, bounding the Jacobian's magnitude provides a regularization mechanism to achieve smoother surfaces Varga et al. (2018). Hoffman et al. (2019) found the decision landscapes of Jacobian regularized classifiers to change less abruptly, with smoother boundaries increasing resilience to adversarial attacks. Their algorithm constrains hidden representations in favor of differential smoothness and robustness efficiently.

In summary, it appears that a representation's geometry, its spectrum and robustness are closely interrelated. Although some links are well understood, in general their mutual dependence remains unclear. Since Krotov and Hopfield's learning rule is directly related to the physiological learning processes, to test whether it reproduces Stringer's spectral property seems obvious. Being a mechanistic model makes it in principle an ideal candidate device for future understanding these connections, because its connectivity dynamics are explicitly stated. In this paper, we study the cross relations between the power spectral decay, geometric properties, robustness and expressivity (model performance). At first, we test Krotov and Hopfield's model for adversarial robustness

with respect to random corruption and white box attacks. Based on the results in Patel & Kozma (2020), we expect it to be more robust compared to end-to-end backpropagation trained models. In an attempt to understand the underlying mechanism, we probe the representation's properties. At first, we examine whether the model reproduces spectral decays conforming to a power law. Using the mentioned regularization methods allows to specifically constrain the optimization for smooth or power law compliant representations. With this, we study the mutual implications of the latent characteristics in a systematic manner.

## 2 METHODS

**Architectural choices**  To study the implications of structural properties in the hidden representation, we assess relative model performance and constrain the structure of our neural network model to an Encoder-Decoder architecture. This choice justifies simpler function classes to control parameters and limit effects due to large model complexity. Because the regularizers are architecture-agnostic beyond the existence of a hidden layer, compatibility concerning Krotov & Hopfield's hybrid model is the limiting factor. Consequently, we choose a Multi-Layer Perceptron model, similar to that in Krotov & Hopfield (2019)

$$\boldsymbol{h}(\boldsymbol{x}) = \boldsymbol{W}\,\boldsymbol{x}$$
$$\hat{\boldsymbol{h}}(\boldsymbol{h}) = \mathrm{ReLU}(\boldsymbol{h})^n \tag{1}$$
$$\boldsymbol{y}(\hat{\boldsymbol{h}}) = \boldsymbol{A}\,\hat{\boldsymbol{h}} + \boldsymbol{b}\quad.$$

Here, $\boldsymbol{x}$ denotes a flattened single image.

Stringer's theory makes statements about the functional relationship between eigenvalues and their index in the ordered spectrum of principal components in the set of the model's representations. Because principal components correspond to the eigenvectors of the respective covariance matrix $\mathrm{Cov}(h, h)$, we study the covariance spectrum $\{\lambda_n\}_{1 \le n \le N}$ in descending order: $\lambda_1 \ge \lambda_2 \ge \cdots \ge \lambda_N$. According to Stringer et al. (2019), optimal encodings follow a power law

$$\lambda_n = \lambda_1\,n^{-\alpha} \tag{2}$$

with an exponent $\alpha > 1$. To detect them, we make use of their scale invariance property. Changing the dimensionality of the underlying representations, real power laws will stay the same if they are correctly normalized. We restrict our quantitative analysis of the power law to finding the exponent $\alpha$ using linear regression on the double logarithmic representation of the power law. Because we expect to see boundary effects, we will do this analysis in regions away from the boundary. We justify these choices in greater detail in Subsection A.3 of the Appendix.

In Krotov & Hopfield (2019), the authors suggest a biologically inspired dynamic learning rule to learn the latent representations of an image classifier in an unsupervised scheme. The decoder of the classifier is learned in a supervised manner using backpropagation. For $\boldsymbol{S}$ denoting the synaptic adjacency matrix between the input and the latent layer, a forward pass in Krotov and Hopfield's encoder translates to

$$h_i(\boldsymbol{x}) = \sum_j S_{i,j}\,|S_{i,j}|^{p-2}\,x_j \tag{3}$$

in index notation. Identifying the weight matrix $W_{i,j}$ with $S_{i,j}\,|S_{i,j}|^{p-2}$, we formally recover the first mapping in Equation 1.

**Unsupervised training**  Krotov and Hopfield's synaptic updating rule was derived implementing renowned neuroplastic mechanisms. A batch parallel approximation of it reads

$$\Delta S_{i,j} = \lambda_L\,\mathbb{E}_{\boldsymbol{x} \in \mathcal{B}}\left[g\left(h_i(\boldsymbol{x})\right)\left(x_j - h_i(\boldsymbol{x})\,S_{i,j}\right)\right] \tag{4}$$

with

$$g(h_i) = \begin{cases} 1 & \Leftrightarrow & h_i = \max\left[\boldsymbol{h}\right] \\ -\delta & \Leftrightarrow & h_i = \max^k\left[\boldsymbol{h}\right] \\ 0 & \text{else} \end{cases} \tag{5}$$

realizing a Winner-Take-All as well as an inhibition mechanism. Here, $\max^k\left[\cdot\right]$ denotes the $k$-th maximum in the set of entries of the vector $h$. We choose the values of hyperparameters consistently

with Krotov & Hopfield (2019). Under these conditions, the model learns prototypic representations and some other features of the data set. However, we also notice that some synapses do not converge. Because the are more noisy, they account for higher variance contributions to the representation's spectrum. After pruning their distribution by ablating all synaptic connections that account for variances above $0.0015$, model performance remains unaffected. To avoid potential side effects, we therefore decided to go with the ablated synapses. For a detailed description of the procedure and its effects on the spectrum, we refer to Section A.1.

**Supervised learning, loss and regularizations** All supervised training, which includes learning only the decoder in Krotov and Hopfield's hybrid model and all weights in the Multi Layer Perceptron (MLP), was based on optimizing the Cross Entropy Loss using the Adam optimizer and mini batches consisting of $1000$ examples each. Besides the pure MLP, we complemented the total loss by adding regularization terms to achieve desired properties in the hidden layer. To study the implications of smoother encodings, we used L2 and Hoffman et al. (2019)'s Jacobian regularization with $n_{\mathrm{proj}} = 3$ projections, but only on hidden representation states. Moreover, we spectrally regularizered the hidden representations for a $n^{-1}$ spectrum using Nassar et al. (2020)'s method. For the results in this paper, we used the CIFAR10 datasets for training and testing.

In terms of structural and spectral properties of the hidden, or rather latent, representations, we expect L2 and Jacobian regularization to have the same effect. In Subsection A.4, we lay down our argument more comprehensively. However, since we measure robustness with regard to the model's prediction, which involves decoding of the hidden representation, they might affect our robustness measures differently. We suspect such also because local and global bounding might affect decoding differently.

**Perturbation experiments** We tested the model robustness against random perturbations and adversarial attacks. In the case of random perturbations, we drew random unit vectors for each image in the model's input space to perturb the image in that direction with magnitude $\epsilon$. These random vectors were created by sampling entries from a standard normal distribution and normalizing the length of the vector to unity. Next to random perturbation, we tested robustness against adversarial attacks, in particular the Fast Gradient Sign Method (FSGM) Goodfellow et al. (2015) and Projected Gradient Descent (PGD) Kurakin et al. (2017); Madry et al. (2019) using $N = 10$ iterations.

To measure model performance under these perturbations, we captured two metrics: relative accuracy and critical distance. Because each model achieves a different prediction accuracy score, and were not interested in the robustness towards false predictions, we normalized the test data for correct predictions for each model to have a common ground to compare them amongst each other. In consequence, all models exhibit $100\%$ accuracy in predictions on their individual test set without perturbation ($\epsilon = 0$). Amongst all adversarial methods, we have varied the perturbation strength $\epsilon$ uniformly on a logarithmic scale until the relative accuracy saturated. Plotting relative accuracy against $\epsilon$ yields smooth curves, whereby a faster or steeper drop in relative accuracy corresponds to a less resilient classifier. Next to this set-wide measure, we recorded the minimal fooling, or critical, distance $\|\Delta \mathbf{x}\|_{\mathrm{crit}}$ for each image in the individual test set and each classifier. This measure is the euclidean length of the perturbation vector in the input image space when the corresponding image is just misclassified. The resulting distributions of critical distances and their statistics provide additional information with respect to the nature of resilience on the individual image level.

To get a qualitative impression on the representational geometry, we examine the model's decision landscape on a randomly projected plane in input space as in Hoffman et al. (2019). In addition, we also visualize the map of the hidden representation's Jacobian norm on the same plane which provides quantitative information about the local change of the encoding and and impression about its relative change, reflecting the curvature of the surface.

## 3 RESULTS

**Comparison in adversarial robustness** We tested the robustness of Krotov and Hopfield's hybrid model (KH) against random perturbations in the input as well as FGSM and PGD adversarial attacks in comparison with end-to-end backpropagation trained models. Next to a naive end-to-end model (BP), we tested L2, Jacobian (JReg, Hoffman et al. (2019)) and spectral regularization (SpecReg, Nassar et al. (2020)). The results of our perturbation experiments are shown in Figure 1. Left hand

side panels show relative accuracy as a function of the perturbation parameter $\epsilon$. Additionally, the right hand side panels visualize the distributions of critical (minimum fooling) distances. Across all attacks and measures, the hybrid model outperforms the others regarding robustness, followed by L2 and Jacobian regularization in that order. In terms of FGSM and PGD resilience, the naive and the spectrally regularized model perform similarly bad. However, concerning random perturbations, spectral regularization evidently yields worse results. Generally across models, the order of severity is: random perturbations, FGSM and PGD. Thus, robustness declines according to how specifically targeted the attack is. As expected, relative accuracy, mean and median critical distance contain the same information. Interestingly, however, the mean and the variance in critical distance appear to be directly correlated. This means that with more resilient models, selected images are also less coherently correctly classified. Consequently, the least robust models are so most coherently.

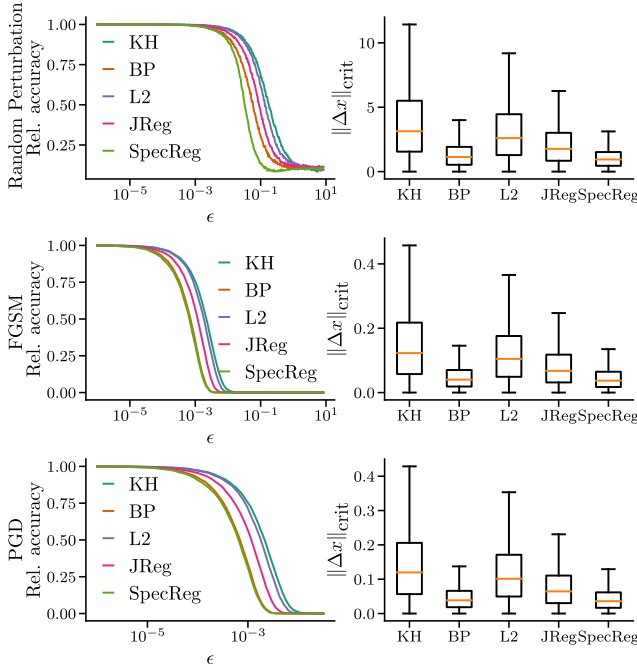

Figure 1: **Left panel:** Relative accuracy as a function of the perturbation parameter $\epsilon$ for all models of consideration under the three adversarial attacks. **Right panel:** Distributions of critical perturbation magnitude $\|\Delta \mathbf{x}\|_{\text{crit.}}$ as L2 distance in the input space (minimal fooling distance) across all images in the input set that were originally correctly classified for all models and attacks considered.

**Robustness-accuracy trade-off** We have studied the relation between robustness and accuracy in terms of test accuracy and median critical fooling distance $\|\Delta \mathbf{x}\|_{\text{crit.}}$ for all five models. Examplarily, we present results for random perturbation and PGD experiments in Figure 6 of Subsection A.2 in the appendix. Although being the most robust, the hybrid model (KH) is also the least accurate. The opposite applies to the naive model (BP). Weight regularization in general finds a better balance compared to the other methods. In particular, local weight regularization (JReg) achieves more robust representations without substantial decrease in accuracy, whereas L2 clearly favors robustness. Here, spectral regularization exhibits the worst results in both measures in compared to the other models.

So far, we have observed a general trend between geometric regularization (L2, JReg) and robustness compared to the hybrid and the naive model. To categorize these results, we will study the mutual implications of geometric and spectral properties next.

**Local learning yields optimal power law representations** Figure 2 shows the ordered and normalized covariance spectra of the data, the Krotov-Hopfield layer right after initialization and after training on CIFAR10 on double logarithmic axes. The spectra are simultaneously shown at different

scales to examine their scaling behavior. Our control, the white noise signal ($\xi(t)$), reveals a flat spectrum as expected besides the final fall due to the finite extensions of the model and data. Also the latent representations of white noise just after initialization are flat. At larger scales, the drop shifts towards the right, but the general flat profile of the spectrum is not affected by the scaling. In turn, we can use the white noise spectra calibrate our analysis of other spectra. For example, because the spectrum in the region appears completely flat, it is reasonable to assume a highconfidence in spectrum profiles anywhere below $n \simeq 500$. In addition, we note that linear regression yields exponents different from 0 contrary to the real value. Therefore, we take the magnitude of this deviation as a proxy for the error of the estimate for $\alpha$.

CIFAR10 test images themselves do not exhibit a power law. Neither does the spectrum's profile appear linear, nor the spectral relation scale independent. However, selecting sub-patches to scale the input dimension might have affected the integrity of the spectrum. Surprisingly, the untrained network appears to have a scale free spectrum, even for CIFAR10 input.

The essential result of our spectral analysis is that the latent representations of the trained encoder consistently exhibit a scale free power law spectrum in the region $n < 800$. Scaling only leads to earlier or later drops in the profile, but does not affect its overall shape. Moreover, we notice a slight bump in CIFAR10 spectra for $n > 800$ dimensions which we account to finite boundary effects. Surprisingly, even random signals appear to get projected to power law representations. In general, we notice that the estimated exponents of representations are always larger than those of the pure data. Moreover, exponents related to the CIFAR10 signal are larger than those related to the white noise source.

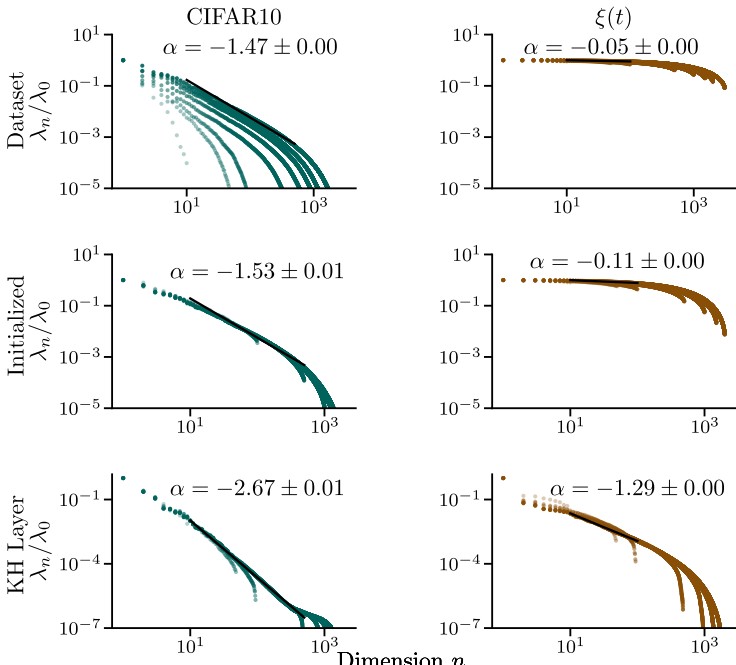

Figure 2: Normalized covariance (PCA) spectra of latent representations $\mathbf{h}(\mathbf{x})$ across CIFAR10 and Gaussian white noise $\xi(t)$ random input. The displayed spectra are those of the signals themselves, the hybrid model suggested by Krotov & Hopfield (2019) after initialization (Initialized) and after unsupervised training (KH Layer).

**The link between geometry, spectrum and performance**   Figure 3 summarizes the ordered covariance spectra of all end-to-end backpropagation trained models for CIFAR10 as well as Gaussian white noise input. At first, we note that the naive Single Hidden Layer Perceptron (SHLP) model does not have a power spectrum, neither in terms of its profile shape nor its scaling. In MLPs, the white noise spectrum seems to have a particularly complex profile. L2 and Jacobian regularization appear to produce qualitatively similar spectra with steeper decay slopes than the naive model.

With respect to that, L2 spectrum exhibits an even larger exponent estimate than that of the Jacobian regularizer. It is interesting that weight regularization appears to cause a similar bump in the spectrum towards higher $n$ as in the KH Layer plot in Figure 2. The fact that this bump is more pronounced in the steeper L2 spectrum suggests that this might a result of higher compression in the more dominant components. As expected, spectral regularization achieves a good power law spectrum for CIFAR10 input. In terms of quality, it is comparable to that of the hybrid model in Figure 2. However, since the model was optimized for $\alpha = 1$, its estimated exponent including the inaccuracy deviates more from the target than expected. Moreover, we see that for white noise input, the spectrally regularized model exhibits a completely flat spectrum, similar to the flat white noise spectra. On closer inspection, we notice that both spectra resemble those of the untrained encoder in Figure 2. Overall, we observe that optimizing for smoothness in addition to accuracy does not seem sufficient to enforce power law spectra. Additionally, the power law spectrum in SpecReg does not generalize to arbitrary inputs in contrast to the KH Layer.

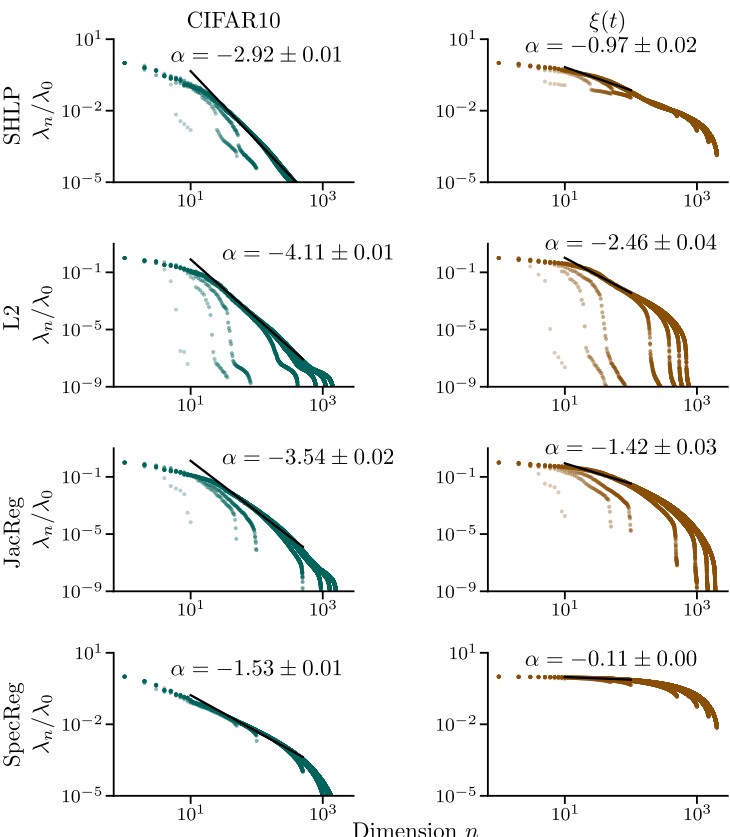

Figure 3: Normalized covariance (PCA) spectra of latent representations $\mathbf{h}(x)$ across CIFAR10 and Gaussian white noise $\xi(t)$ random input. The displayed spectra are those of the fully gradient optimised models without (SHLP) and with regularisation (L2, JacReg, SpecReg).

As in Hoffman et al. (2019), we plotted an exemplary decision boundary landscape of a common random plane projection in the model's input space. The resulting decision landscapes of all models are shown in the lower panel of Figure 4. There, we started with an example image that was correctly classified across all models (center of the plots) and tracked the model's decisions along with its confidence to estimate the decision landscape for a linear continuation around the original image in a random two-dimensional plane. In addition, we also visualize the Frobenius norm of the exact Jacobian of the latent activations $\hat{\mathbf{h}}$ of each model for the same random projection, as a measure of its smoothness. With it, we gain two pieces of information within the plane. The value of the norm serves as a local estimate of the encoding manifold's change. Moreover, the relative change of the value of the norm in space provides a qualitative estimate of its curvature. Thus, in total we

get qualitative information concerning the latent and hidden representation's roughness, and how this affects the model's predictions, out of this plot. In direct comparison, we observe that weight

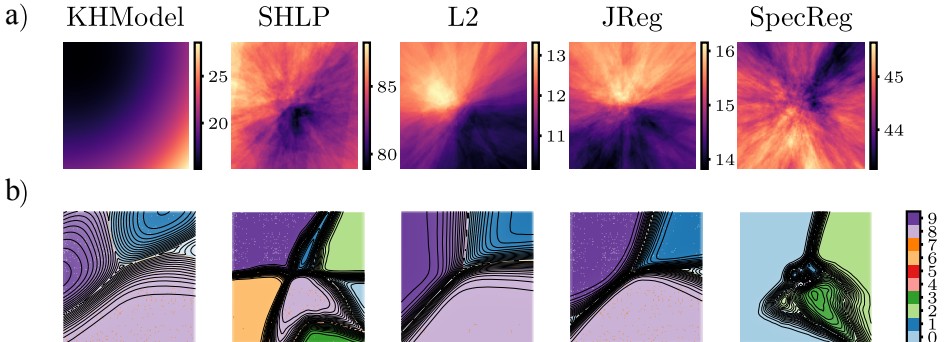

Figure 4: Norm of the Jacobian of latent activations $\hat{h}$ (upper panel) and decision landscape (lower panel) of studied models. Similarly to Hoffman et al. (2019), the displayed planes are created by grid sampling from a random plane projection in the model's input space.

regularization, in general, results in the lowest norm values, whereby L2 regularization achieves the lowest. Also, these low values in panel a), indicating comparably small changes in the representation surface, translate into smooth decision boundaries with very clear borders in panel b). In contrast to that, the naive model exhibits by far the highest Jacobian scores which translate into rather small decision domains with rougher edges. Regarding the norm values, spectral regularization resides between those poles and shows rather large decision domain patches. However, especially in the close neighborhood of the center, which represents the starting image, the decision terrain appears particularly rough. In terms of the range of the Jacobian, Krotov & Hopfield (2019)'s model (KHModel) is closest to weight regularization which is also reflected in its decision landscape. In fact, the general structure of the decision lanscape between these models is almost identical, besides the orientation of the gradient. However, the hybrid model's confidence changes less abruptly between decision domains, compared to the weight regularized models.

In terms of relative change of the Jacobian norm across the field, there are essential differences between local learning and backpropagation. We observe that the norm value changes only gradually and smoothly along the latent activation landscape of the KH model. Moreover, the Jacobian score changes linearly from smaller to higher values starting in the upper left corner and ending in the lower right one. These quantitative results indicate the highest curvature of the latent representation to occur perpendicular to the gradient which coincides with the major trench in the corresponding decision landscape. From this, we see that the major trend in the geometry of the model's latent representation essentially informs the model's decision. The same analysis yields similar results in case of L2 and Jacobian regularization. However, we observe that the relative changes in the hidden Jacobian landscapes, which is reflected in the image's contrast, are less smooth which suggests a rougher surface. Eventually, the respective relative changes in the naive and spectrally regularized models appear similarly abrupt and show no trend, particularly in the case of spectral regularization.

## 4 DISCUSSION

In this paper, we studied the mutual implications between geometric and spectral properties of latent and hidden representations and how they affect performance of simple two layer perceptron models on the basis of Krotov and Hopfield's local learning model. This theme resonates with the exploration undertaken in Wang & Ponce (2021), where the structural intricacies of GANs' latent spaces and their spectral properties are analyzed, elucidating their influence on image generation. We measured performance in terms of accuracy and robustness against random perturbations, FGSM as well as PGD attacks. In general, robustness and accuracy are consistently negatively correlated across models apart from spectral regularization, where the hybrid model was most robust but least accurate. To understand why, we studied the smoothness of the model's representation manifold in terms of the Jacobian as well as its covariance spectrum with regard to power law profiles. Both are

established mechanisms to achieve general model robustness against data corruption. To establish a baseline for comparison, we also studied the regularizers that optimize for the respective presumably optimal properties. The local learning model exhibits both, a comparably smooth representation surface as well as a power law spectrum, indicating presumably optimally balanced representations.

Krotov and Hopfield's model yields latent representation manifolds similarly smooth as with weight regularization. Providing additional results on white box attacks, we find that smoother representation manifolds result in more resilient models in agreement with Krotov & Hopfield (2016; 2019); Grinberg et al. (2019); Hoffman et al. (2019). By comparison with their decision landscapes, we see that geometric properties in hidden representations translate down to geometric properties in the classification layer, although they were not explicitly regularized. With this, smoother hidden representations yield smoother decision boundaries, thus increasing robustness of the classifier overall. This finding is in line with Zavatone-Veth et al. (2023), who explore how training induces geometric transformations in neural networks, particularly magnifying areas near decision boundaries, which significantly impacts class differentiation and network robustness. Besides the local learning model, L2 as well as Jacobian regularization constitute the most promising approaches of those studied to achieve high resiliency. Although they were optimized for accuracy and smoothness simultaneously, neither of the weight regularized models exhibit spectra close to a power law. Consequently, we conclude that either both models are located afar from the optimum in parameter space. Moreover, an ideal balance might not be a sufficient criterion for this class of models.

Controlling the spectrum directly had almost no implications, neither regarding geometry nor performance. Following Nassar et al. (2020), we would have expected an increase in robustness from spectral regularization that we did not reproduce. We observe that spectral regularization decreases the magnitude of the latent Jacobian compared to the naive representation but does not benefit robustness This can be explained by a stronger folded surface, which the abrupt changes in the Jacobian norm hint at. In contrast to local learning, the regularized power spectrum does not generalize to white noise data. Consequently, the hybrid model constitutes the more interesting case to study. Its latent spectrum falls more quickly than the dataset's which can be seen from the estimated exponents. If we assume them to reflect the intrinsic (fractal) dimension of the signal, even when the spectra do not follow power laws, we qualitatively confirm Stringer et al. (2019) in that optimal representations have higher exponents corresponding to lower dimensions. However, representations are also generally expected to are lower dimensional than the original data because they formally constitute some form of data compression. In this light, also the estimated exponents of the end-to-end backpropagated models are consistent. For example, the decay of the weight regularized models is steeper in the estimation regions compared to the naive model since constraint (smoother) representations lead to higher degrees of compression. In particular, the L2 spectrum is characterized by an even steeper fall in comparison to Jacobian regularization. As predicted by Stringer et al. (2019), the spectral decay is generally stimulus dependent, with flatter spectra reflecting higher dimensional data.

We notice that there remain gaps that are not explained by the current state of the theory. Partly, they might be a result of our model's finite nature whereas Stringer's arguments rely on properties in infinitely dimensional Hilbert spaces. In any case, our results suggest that many open questions remain regarding understanding robustness of classifiers, even for simple function classes.

To close these gaps, our discovery, that Krotov and Hopfield's local learning rule yields robust representations that perform well, are smooth and exhibit a close to ideal power law spectrum, might be of significant impact for upcoming studies. With properties that match with the ideal model in Stringer's theory it is a promising mechanistic study case. Moreover, our work could provide a starting point towards understanding how power law spectra determine optimally smooth encodings and beyond. Because of the model's biological foundations, our results also provide insights into how robust neural networks are mechanistically realized in the mammalian brain, and how they can be achieved in artificial systems.

**Reproducibility** Our results were generated using the methods we mention with parameters according to their references. Whenever our parameter values differ from that in the resources we explicitly state our values in the text.

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

# A  APPENDIX

## A.1  POSTPROCESSING AND PRUNING

Figure 5 a) shows a selected subset of synaptic weights in $S$ after unsupervised training. Each image block resembles the weight values $S_{i,.}$ that linearly project the input onto one of the hidden units. As in Krotov & Hopfield (2019), we display them in the same format as the original input, in this case in the same format as CIFAR10 images. The model has mostly learned prototypic representations and other features of the data set in agreement with Krotov & Hopfield (2019). However, we also notice that some of the image blocks in Figure 5 a) Raw do not seem to have any correlations: they appear to be random. We suspect that these subsets of synapses do not encode any information, but rather constitute an artifact due to the finite amount of training, and would have converged otherwise. Appearing more noisy, they account for higher variance contributions to the representations, which is reflected in their covariance spectra. Figure 5 b) shows the distribution of per-image variances. Next to the major mode of the distribution located just below 0.001, we see another, less pronounced, minor mode at 0.002. Assuming these represent the allegedly not converged subset, we prune the distribution by ablating all synaptic connections that account for variances above 0.0015. In agreement with this assumption, we notice that the noisy subsets have vanished in Figure 5 a) Pruned. Moreover, Figure 5 c) shows the latent representation's ordered covariance spectra of both, with an without ablation, in a double logarithmic plot. On the left panel, the model was subject to images from the CIFAR10 testing dataset, whereas on the right hand side, results are shown for random white noise input. Generally, we see that the representation's spectra appear linear, apart from the left and the right boundaries, in all cases which suggests them to follow a power law. As discussed earlier, this alleged linearity is not sufficient to identify the relation as a power law. A comprehensive analysis follows in the results section. In particular, the noise-input spectrum of the untouched model (Raw) is characterized by a substantial drop after the first two eigenvalues. On closer inspection, one identifies a similar characteristic in the profile of the corresponding CIFAR10 spectrum. The fact that it no longer appears in the ablated spectra confirms our suspicion that it was caused by high variance contributions in the set of synapses $S$. Because the noisy image synapses do not appear to encode meaningful information, but the ablated spectra show a more coherent putative power law, we chose the ablated model for further investigations.

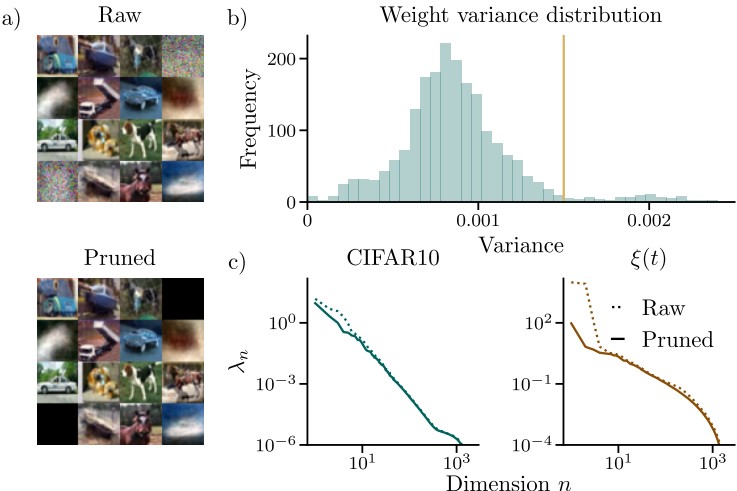

Figure 5: Synapse and representation characteristics after unsupervised training with the local learning rule. **a)**: Synaptic connections presented as images. Some entry blocks appear to resemble noise (Raw). We prune noisy contributions to improve spectral properties (Pruned). **b)**: Distribution of total variances per image block in $S$. The distribution is bimodal with the major mode centered below $0.001$ and the minor mode around $0.002$. Pruning the higher variance contributions by setting a cutoff threshold at $0.015$ ablates noisy images in $S$ (see panel **a)**), and eliminates the initial drop in the representation's covariance spectrum (see panel **b)**). The latter results in comparably clean power relations between the eigenvalues of the first components. **c)**: Ordered covariance spectra of the representations corresponding to both, the original (Raw) and ablated (Pruned) set of synapses, subjected to CIFAR10 (left panel) or Gaussian noise (right panel).

## A.2 ROBUSTNESS-ACCURACY TRADE-OFF

Figure 6 shows the relation between test accuracy and median critical fooling distance $\|\Delta \mathbf{x}\|_{\text{crit.}}$ of all five models. As an example, we present results for a) random perturbations and b) PGD. Aside from spectral regularization, robustness and accuracy are negatively correlated.

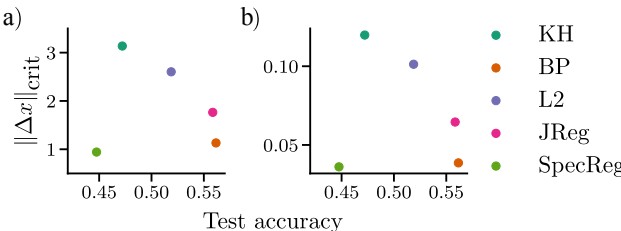

Figure 6: Relational plot between test accuracy and the median of critical distances $\|\Delta \mathbf{x}\|_{\text{crit.}}$ across models as a measure of robustness regarding **a)**: random perturbations and **b)**: Projected Gradient Descent.

## A.3 POWER LAWS

Beyond indicating putative optimal encodings, power laws are special because of their scale invariance. For any fixed exponent, scaling transformations $n \mapsto a\,n$ leave the power law property unaffected

$$\lambda_n \mapsto \lambda_{a\,n} = \lambda_1 \,(a\,n)^{-\alpha} = a^{-\alpha}\,\lambda_n \quad . \tag{6}$$

From that, we immediately see that $\lambda_{a\,1} = a^{-\alpha}\lambda_1$. Consequently, a power law in $\lambda_n$ can be detected as a power law in

$$\tilde{\lambda}_n = \frac{\lambda_n}{\lambda_1} = n^{-\beta} \quad , \tag{7}$$

the difference is that the functional relationship in $\tilde{\lambda}_n$ is independent of the absolute magnitude of eigenvalues or scale and is therefore comparable between different models at arbitrary scales.

We restrict our quantitative analysis of the power law to identifying the exponent $\alpha$. At this point we stress the important difference between our notion of a power law and what is widely perceived. Primarily, power laws refer to probability distributions or their density functions. In those cases, it is best to estimate parameters leveraging the vast amounts of statistical methods that exist. Here, however, the term power law refers to a functional relationship. Methods for parameter estimation in functional relationships usually come down to regression methods. In a double logarithmic plot, power laws appear as linear relations

$$\log \tilde{\lambda}_n = -\alpha \, \log n \quad , \tag{8}$$

with former exponents corresponding to slopes. Therefore, we use linear regression to identify $\alpha$ and its error in the double logarithmic representation of the ordered spectrum. Because statistical tests for linear regression focus on monotonicity hypotheses, which are trivially met in ordered spectra, we renounce analyses regarding our estimates' significance.

## A.4 SIMILARITY OF L2 AND JACOBIAN REGULARIZATION

In terms of spectral implications, we should expect weight based regularization methods to have a similar effect, whether it is Jacobian or L2 regularization. The notions of robustness and continuity are closely related in the context of static models or functions. Therefore, Stringer's statement, smoother representations account for more robust models, intuitively makes sense. With robustness, we are generally interested in how comparably small perturbations in the input $x \mapsto x + p$ locally translate into changes in the output. With Stringer et al. (2019) in mind, we are particularly interested in changes in the latent or hidden representation $\hat{h}(x) \mapsto \hat{h}(x + p)$. Assuming local differentiability allows us to locally quantify how perturbations translate by applying Taylor expansion.

$$\left\| \hat{h}(\boldsymbol{x} + \boldsymbol{p}) - \hat{h}(\boldsymbol{x}) \right\| = \left\| \hat{h} + \frac{\partial \hat{h}(\boldsymbol{x})}{\partial \boldsymbol{x}} \, \boldsymbol{p} + \cdots - \hat{h}(\boldsymbol{x}) \right\|$$
$$\leq \left\| \frac{\partial \hat{h}}{\partial \boldsymbol{x}} \right\| \|\boldsymbol{p}\| + \mathcal{O}(\|\boldsymbol{p}\|^2) \tag{9}$$

Comparably low perturbations in the input scale with a factor of $\|D\hat{h}(x)\|$ in the representations. Thus, bounding the Jacobian yields local stability with respect to small perturbations. Because structurally $\hat{h}(\boldsymbol{x}) = \sigma(\boldsymbol{W}\,\boldsymbol{x})$, the norm of the Jacobian directly relates to the norm of the weights

$$\left\| \frac{\partial \hat{h}}{\partial \boldsymbol{x}} \right\| = \left\| \frac{\partial \sigma(\boldsymbol{W}\,\boldsymbol{x})}{\partial (\boldsymbol{W}\,\boldsymbol{x})} \, \boldsymbol{W} \right\| \quad . \tag{10}$$

Consequently, Jacobian and L2 regularization locally have the same effect on robustness. In contrast to Jacobian regularization, however, L2 regularization bounds the weight matrix globally. Since $\sigma = \text{ReLU}$ in our case, it might not be differentiable at $\boldsymbol{W}\,\boldsymbol{x}$. Moreover, these considerations apply only locally and to the hidden representation and have direct consequences with respect to its geometry. Following Stringer's argument, we therefore expect both forms of weight regularization to have similar effects on its covariance spectrum.

