# OpenReview forum: "Exploring mechanisms of Neural Robustness: probing the bridge between geometry and spectrum"
_ICLR.cc/2024/Conference — Submitted to ICLR 2024_

### Official Review · Reviewer_wEmA · 2023-10-29

**Soundness:** 3 good
**Presentation:** 2 fair
**Contribution:** 3 good
**Rating:** 6
**Confidence:** 4

**Summary:**

The paper analyses the robustness of Krotov-Hopfield networks to noise by analyzing the eigenvalue spectrum of their internal representations. They find that it naturally produces a power-law spectrum, as had been observed in the brain by Stringer et al.

**Strengths:**

This study provides some useful information on the robustness of the Krotov-Hopfield model

**Weaknesses:**

The question of robustness and eigenvalue spectra is certainly interesting, but the focus on the KH model limits the impact of this work.  The study would have been stronger with a more general analysis of this for several learning rules.  For example, the KH model generates its hidden layer in a unsupervised manner.  Do other unsupervised rules act similarly?  There is some analysis of multiple rules in Figure 5, but not much in the way of a general theory, and the paper would have been stronger with more of this.

**Questions:**

There are two papers cited by Krotov and Hopfield, but you don’t always say which one you are referring to.  Please distinguish them by adding a year every time!

The matrix S is not clear.  What does “synaptic adjacency” mean?  Is it defined by W=S ||S||^p-2?   Also the norm ||S|| is not defined.  Do you actually mean absolute value |S|?

\lambda means learning rate in equation (4) but covariance eigenvalue in figure 1c.

What are the multiple curves in Figure 2?

---

### Official Review · Reviewer_74wa · 2023-10-29

**Soundness:** 2 fair
**Presentation:** 1 poor
**Contribution:** 1 poor
**Rating:** 3
**Confidence:** 4

**Summary:**

The authors investigate whether a neural network trained on CIFAR with biologically inspired local learning rules contains activations with a power-law-like spectrum, which has been proposed as a property of biological systems. They compare this network to others trained with spectral regularization and Jacobian regularization, which explicitly induce power-law-like spectra in the network. The adversarial robustness of these networks is also investigated.

**Strengths:**

The authors' experiments bridge many lines of work in machine learning and neuroscience. The question of power-law spectra in biological and artificial neural networks is topical and of interest to neuroscience and machine learning practitioners interested in adversarial robustness.

**Weaknesses:**

1) The paper is difficult to follow. In some places, there are many methodological details presented, but the experiment hypothesis and conclusions need to be better described. In general, the writing could be clearer and more precise.
2) Many of the experiments seem preliminary, which makes it hard to draw conclusions. For instance, the authors state in the discussion “we could not reproduce Nassar et. al. (2020)’s results, probably due to the differences in our modeling choices”, and in the appendix “most of our results suggest that we might have made sub-optimal choices in our methods regarding competitive results with spectral regularizers”. While I appreciate the authors being honest about these limitations, this leaves more questions than answers. What would change if models were made that *did* reproduce these previous results? Is there something fundamentally non-reproducible about the previous work? This makes the presented results difficult to interpret, and in the words of the authors “we ought to be cautious with generalizing them beyond the scope of this work”.
3) There was not enough relation to previous literature. For instance, the authors motivate adversarial robustness with an example from cancer diagnosis, but there are many papers in the NeuroAI world that have directly looked at the robustness of models compared to human observers. This seems like it would be more relevant to the neuroscience audience that this paper is targeted towards. I encourage the authors to look at this work (some places to start would be Geirhos et al. 2018, Geirhos et al. 2021, Guo et al 2022, Harrington et al. 2022, Feather et al. 2023 but there are many many more).
4) Minor: The citation format in the text seems to be incorrect with missing “()” around the text citations.
5) Minor: In the first two sentences of section 3.2 the authors state that Figure 3 shows the spectrum of the KH layer, but this is not in Figure 3 (although it is in Figure 2).

**Questions:**

a) The authors introduce the acronym “BNN” and refer to these as “models.” When the authors refer to “BNNs” are they talking about true biological systems or models of biological systems? “BNN” is not a common acronym to my knowledge.

b) How do the authors handle the fitting for the “alpha” value? In Stringer et al. I believe that the first few and last few eigenvalues are ignored. How does that play into the fitting here?

c) Which points (scales) are used for fitting the alpha value in Figure 2? Additionally, the different colored dots are not defined in the plot. A legend should be included.

d) It would be helpful to show the non-normalized robustness plots. Because the clean performance significantly differs these are hard to interpret as shown.

e) Figure 4 is interesting, however the statements about “smoother” decision landscapes should be quantified if one is to directly compare the results to the adversarial robustness curves.

---

### Official Review · Reviewer_1KBf · 2023-10-30

**Soundness:** 2 fair
**Presentation:** 2 fair
**Contribution:** 2 fair
**Rating:** 3
**Confidence:** 4

**Summary:**

This paper seeks to investigate the relationship between the smoothness of the representation and the spectra of its covariance matrix.

I found the paper to be not particularly well organized, and somewhat difficult to follow. While the paper described a series of analyses, these analyses are overall preliminary. There is not a clear key result emerging from these analyses that would fundamentally change our understanding of the problem. In sum, I feel that while the question studied in the paper is potentially interesting, the results need to be substantially strengthened.

**Strengths:**

— Understanding the robustness of the representation is an important problem.
— The attempt to leverage insights from neuroscience to improve machine learning algorithms should be applauded.

**Weaknesses:**

- The main problem that I see with the current version of the paper is that there isn’t a clear and impactful result reported in the paper.

- The presentation is not particularly well organized and needs major improvements. The descriptions of the analyses and results are messy.

- Various statements are not rigorous. For example, the paper stated, “biological neurons adjust connectivity based on neighboring cell activity”. What is the evidence for this? Also, the brain has abundant long-range connections.

- The paper is motivated by Stringer et al (Nature, 2019). However, the interpretation of their results seems to be somewhat misleading. The paper stated “They prove that this smoothness is linked to the functional form of a power law decay in the manifold’s covariance spectrum. In particular, balancing accuracy and robustness, a sweet spot lies close to a 1/n power law, with n denoting the index of the ordered spectral component. ” But I don’t think Stringer et al 2019 proved a connection between the 1/n spectrum with the robustness. It was a conjecture described in the last paragraph of that paper. Also, Stringer et al predict different power-law relationships depending on the properties of the stimulus set (e.g., natural scene v.s. gratings). It is unclear whether the current results are consistent with theirs.

**Questions:**

-- It would be helpful if the paper could be systematically revised to make the analysis more systematic and the key points more clear.

-- What is the point of the left panel of Fig 1c? The pruned version mainly differed from the Raw in the power of the first few components while the scaling-law concerns the properties of the tails. So I am not sure why this difference would matter for the scaling-law.

-- The analysis in Fig 4 is interesting. But it’s just one example. Would there be a systematic way to quantify this?

---

### Official Review · Reviewer_8ibB · 2023-10-31

**Soundness:** 3 good
**Presentation:** 1 poor
**Contribution:** 2 fair
**Rating:** 5
**Confidence:** 4

**Summary:**

### Summary

The authors conducted a series of carefully designed analyses on 2-layer NN models trained by hopfield-like local learning rules vs backpropagation with regularization, and they examined their representation spectra, local manifold geometry, decision boundary and robustness to white box attack. Through the analyses, they found the networks learned through local biologically plausible learning rules are indeed robust and have "optimal" spectra, while some other models trained via backprop can also achieve comparable robustness without the 1/n spectra, questioning the necessity of this 1/n spectral feature.

**Strengths:**

### Strength

- The detailed dissections of these models are ****applaudable.****
- Providing evidence that the 1/n spectrum of the representation is not necessary for model robustness, (though the evidence is relatively few data points) .
- The latent geometry analysis and its relation to decision boundary is illuminating, esp. by showing the similarity of KH model and L2 and Jacobian regularization model.
- Figures are well made, clean and easy to understood.

**Weaknesses:**

### Weakness

- The biggest issue maybe the paper at its current stage is kind of explorative data analysis in many directions—— so I’m not sure about the central contribution and conclusion. The 1/n spectrum, the geometry of represntation and decision boundary and adversarial robustness are all interesting, but currently they have not connect to form an overall story. It will be better to form a few central claims and then back it up with experiments and data.
- The writing and overall organization of the paper could be improved…
    - In the method section, it will benefit from a more streamlined organization. Currently it’s jumping between method, results and some interpretations…. Some paragraph (Figure 1 and its interpretation) should go to the results section. Adding headings e.g. paragraph titles can also help. e.g. model architecture, unsupervised training, supervised training, spectral analysis etc.
    - Similarly in the result section, better headings for the experiments could be added to guide the readers.
- Generally the paper’s experiment is relatively small scale, majorly with 2 layer MLPs. The generalization to deeper networks or CNNs is desirable for bigger impact.
- For the decision boundary and  analysis, it may be worthwhile to refer to some previous works.
    - For example, the Frob norm of jacobian plot vs the decision boundary is interesting, and [1] devotes the whole paper to understand this. The technical difference is they plotted the volume element, which is the sum of log eigenvalues, instead of sum of squared eigenvalues in your case.
    - Though outside image classification, for GAN networks, [2] analyzed the homogeneity / flatness of the Jacobian eigenframe across the space. which maybe useful for quantification in your case.

[1]: ****Neural networks learn to magnify areas near decision boundaries****  https://arxiv.org/abs/2301.11375

[2]: ****The Geometry of Deep Generative Image Models and its Applications**** https://arxiv.org/abs/2101.06006

### Minor Weakness and typos

- “*Because principal components correspond to the eigenvectors of the respective*
*covariance matrix Cov(h,h)*” should it be $\hat{h},\hat{h}$ ?
- In Figure 2 and Figure 3, what does different color shades mean? is it different runs of the model?
- The notation in Eq. 9 in appendix is not clear, $D$ seems to mean differential operator, but it’s not standard and not defined anywhere, using partial differential or more standard notation may be easier for reader.
- Eq. 10 in appendix is also confusing or even wrong. $\sigma’(Wx)$ seems to be a vector, how do you take the absolute value and time it with the L2 norm of $W$? Are you using some special properties of $\sigma$?
- What is *SHLP?*

**Questions:**

### Questions

- The decision landscape analysis is super interesting, but also quite local and qualitative, do you have more quantitative population statistics for them?
- **Comments:**
Covariance spectrum is a quite global measure of representation; on the other hand, the jacobian norm and adversarial property are super local property [3], so it makes sense that controlling the spectrum is not enough for getting adversrial robust…

[3]: **[Adversarial training is a form of data-dependent operator norm regularization](https://proceedings.neurips.cc/paper/2020/hash/ab7314887865c4265e896c6e209d1cd6-Abstract.html)**

---

### Author Response · Authors · 2023-11-23
**Response and reference to our updated manuscript**

We would like to thank the referees for their valuable feedback. Overall, we share the general critique regarding the structure of the paper, the quality of the writing and the presentation of our contribution. To resolve structural issues, we have revisited our story, restructured the document and rearranged some figures. The most evident changes include
1. move Figure 1. to the appendix  because it is not relevant
2. transfer Figure 6 to the results section
3. Introduce paragraph headings to both, the methods and the results section
Moreover, we went through the document to eradicate typographic and formal errors. We hope the way we changed the story, order of results and arguments clarifies our motivation and benefits to understand our contributions.
In the following, we are going to address each of the reviewers individually to respond to their comments and answer their questions.

### Reviewer 8ibB:
We want to take the opportunity to thank the referee for their positive assessment of our paper as well as their detailed and valuable feedback. We found the literature references helpful und clearly relevant for our research and decided to reference some of them in our text.

Regarding question 1: "The decision landscape analysis is super interesting, but also quite local and qualitative, do you have more quantitative population statistics for them?"
	The objection is justified. We share the concern but do not have respective population statistics, which would allow for a more comprehensive  and quantitative study, yet. We would appreciate any further advice on that.

 Regarding the comment: "Covariance spectrum is a quite global measure of representation; on the other hand, the jacobian norm and adversarial property are super local property [3], so it makes sense that controlling the spectrum is not enough for getting adversrial robust…"
	 We agree with the reviewer's comment. However, the work by Nassar et al. appears to provide constrasting empirical evidence. Yet, we could not reproduce any of the results shown in their paper.

### Reviewer 1KBf:

Thank you for your insightful feedback. We share that our assertions regarding Stringer et al. were incorrect and misleading. In our revised version, we tried to resolve these issues.

Regarding question 2: - "What is the point of the left panel of Fig 1c? The pruned version mainly differed from the Raw in the power of the first few components while the scaling-law concerns the properties of the tails. So I am not sure why this difference would matter for the scaling-law."
	It is correct that we are only interested in the asymptotic behaviour of the spectrum. However, we wanted to check to what extent the high variance synapses contribute to the statistics we study. Since it does not contribute to the story of paper as it stands now, we decided to transfer this Figure and discussion to the appendix.

Regarding question 3: - "The analysis in Fig 4 is interesting. But it’s just one example. Would there be a systematic way to quantify this?""
	You are right. The analysis presented is fairly limited and the question for a systematic quantitative statistic justified. So far, we would think there is but we do not know.

### Reviewer 74wa:

Thank you for the detailed feedback and extensive references to the literature. We were glad the referee provided that starting point for us. We used it to extend our references to the NeuroAI literature in our introduction.

Regarding question b): "How do the authors handle the fitting for the “alpha” value? In Stringer et al. I believe that the first few and last few eigenvalues are ignored. How does that play into the fitting here?"
	Indeed, we also ignore the first and the last few eigenvalues. Based on our spectral analysis with regard to scaling the model, we identify regions in which the profile appears fairly stable unaffected by boundary artefacts.

Regarding question c): "Which points (scales) are used for fitting the alpha value in Figure 2?"
	We use the full (largest) scale for fitting the alpha values.

We would also like to thank the reviewer for their additional comments. We will take them into account for further revisions of our paper.

---

### Author Response · Authors · 2023-11-23
**Individual comment addressing Reviewer wEmA**

### Reviewer wEmA:

Thank you for your insightful comments. We share the concern regarding generality of our results without visiting other unsupervised learning rules. We focussed on the local learning rule mentioned in the paper because it exhibits all properties we refer to (power law spectrum, sufficiently smooth manifold, robustness) which makes it an ideal candidate to study the mechanistic relationships between them. We have not analyzed other learning rules yet and appreciate the recommendation to proceed that way.

Regarding question 1: "There are two papers cited by Krotov and Hopfield, but you don’t always say which one you are referring to. Please distinguish them by adding a year every time!"
	We thank the referee for that notice. We corrected the issue during our revision.

Regarding question 2: "The matrix S is not clear. What does “synaptic adjacency” mean? Is it defined by W=S ||S||^p-2? Also the norm ||S|| is not defined. Do you actually mean absolute value |S|"
	The matrix S is defined according to Krotov & Hopfield (2019). It corresponds to a weight matrix W with regard to the architecture discussed by the relation W=S |S|^p-2 , where |.| denotes the absolute value. It was our mistake that we did write the norm instead of the absolute value there. We have corrected it in our text.

Reharding question 3: "\lambda means learning rate in equation (4) but covariance eigenvalue in figure 1c."
	That is correct, but we try to make the difference clear by the subscript. \lambda_{L} with a capital subscript is reserved for the learning rate whereas \lambda_{i} with lower case subscript refers to an index variable of an index in case the subscript is a numeral.

Regarding question 4: "What are the multiple curves in Figure 2?"
	Because a real power law is characterized by its scale independence, we test for that property by scaling the dimensions of the model (number of latent/hidden neurons). The different lines in Figure 2 refer to the spectra of differently scaled versions of the same model exposed to the same data.

---

### Meta-Review · Area_Chair_pKKQ · 2023-12-08

**Metareview:**

This paper studies 2-layer neural networks trained using either a biologically plausible local learning rule or with backpropagation with different regularizations.  They study the spectra of the learned representations and show the local learning rule leads to a 1/n power law which is conjectured to lead to a smoother representation manifold and  better robustness.  They compare this to networks trained with backprop with L2, jacobian, or spectral regularization and investigate the representation spectrum, smoothness of the manifold, decision boundaries, and robustness of adversarial attacks across these methods.

In general, reviewers found the study interesting and worthwhile.  The findings and analysis and intriguing, for example, that 1/n spectrum is not necessary for robustness.  The connection between ML and neuroscience in terms of neural representation spectrum is an interesting direction.  However, there was wide agreement the presentation of the paper was problematic, the paper was hard to follow, and there was no central story and clear main contributions.  Moreover, the results appear preliminary and concern only small networks.  The consensus is the paper is not ready for publication and can benefit from substantial revision and more experiments.

**Justification For Why Not Higher Score:**

- paper presentation
- small scale results
- problematic interpretation of prior work and lack of connection to other prior work

**Justification For Why Not Lower Score:**

N/A

---

### Decision · Program_Chairs · 2024-01-16

Reject